# Unleashing the Potential of the Diffusion Model in Few-shot Semantic Segmentation

**Muzhi Zhu**[1*]    **Yang Liu**[1*]    **Zekai Luo**[1*]    **Chenchen Jing**[1]
**Hao Chen**[1*]    **Guangkai Xu**[1]    **Xinlong Wang**[2]    **Chunhua Shen**[1]

[1]Zhejiang University        [2]Beijing Academy of Artificial Intelligence

## Abstract

The Diffusion Model has not only garnered noteworthy achievements in the realm of image generation but has also demonstrated its potential as an effective pre-training method utilizing unlabeled data. Drawing from the extensive potential unveiled by the Diffusion Model in both semantic correspondence and open vocabulary segmentation, our work initiates an investigation into employing the Latent Diffusion Model for Few-shot Semantic Segmentation. Recently, inspired by the in-context learning ability of large language models, Few-shot Semantic Segmentation has evolved into In-context Segmentation tasks, morphing into a crucial element in assessing generalist segmentation models. In this context, we concentrate on Few-shot Semantic Segmentation, establishing a solid foundation for the future development of a Diffusion-based generalist model for segmentation. Our initial focus lies in understanding how to facilitate interaction between the query image and the support image, resulting in the proposal of a KV fusion method within the self-attention framework. Subsequently, we delve deeper into optimizing the infusion of information from the support mask and simultaneously re-evaluating how to provide reasonable supervision from the query mask. Based on our analysis, we establish a simple and effective framework named DiffewS, maximally retaining the original Latent Diffusion Model's generative framework and effectively utilizing the pre-training prior. Experimental results demonstrate that our method significantly outperforms the previous SOTA models in multiple settings. Our code is released at: `https://github.com/aim-uofa/DiffewS`

## 1    Introduction

The Diffusion Model (DM) has demonstrated powerful capabilities in multiple visual generation tasks, including image generation [1, 2], image editing [3, 4], video generation [5–7], etc. At the same time, DM has also been proven to be a powerful method for self-supervised pre-training [8, 9] employing unlabelled data. To exploit the representation ability of DM, there are currently two emerging topics in vision research: improving the learning paradigm [10, 11] and downstream task adaptation[12–14]. The latter often focuses on the Latent Diffusion Model [2] (LDM). By compressing images into latent space, it significantly decreases computational expenses and emerges as the first open-source Text-to-Image Diffusion Model scaled up to the LAION-5B [15] level. For example, ODISE [16],DVP [12], DatasetDM [17] adapt LDM to multiple tasks such as depth estimation, semantic segmentation, but they all require training additional decoder heads, which increases training costs and may undermine the generalization ability and generation quality. Therefore, some works [13, 14] have emerged that attempt to repurpose the Diffusion Model's generative framework and apply it to visual perception tasks without adding extra decoder heads. Nonetheless, these paradigms still cannot uniformly adapt to all tasks.

---

[*]MZ, YL and ZL contributed equally. YL is the project lead. HC is the corresponding author.

38th Conference on Neural Information Processing Systems (NeurIPS 2024).

Let's reconsider the most fundamental question in using generative models for visual perception: *how to design a fine-tuning framework that can guarantee both generalization ability and precise prediction of details?* Unfortunately, existing methods do not sufficiently address this challenge. The demands of the FSS task for open-set generalization and high-quality segmentation results precisely align with this challenge. Thus, **our first motivation** is to further address the fundamental question posed above by exploring the Diffusion Model on the FSS task.

FSS aims to segment query images given support samples. Traditional FSS methods[18–20] rely on a pre-trained backbone, achieving semantic matching and pixel-level prediction tasks through designing complex modules and long-term training. Recently, with the emergence of SAM [21], some works are based on foundation models to complete FSS, such as Matcher [22]. It employs DINO[23] for semantic matching and SAM for segmentation. Similarly, other works [24, 25] combine SAM with CLIP or MLLM to complete other open-set segmentation tasks. The current methods deal with matching(semantic) and segmentation as two distinct tasks through different modules. The Diffusion Model itself, however, exhibits significant potential in fine-grained pixel prediction tasks[13, 14, 16] and semantic correspondence tasks [26–28]. Hence, we seek to maximize the reuse of the generative framework by taking advantage of the innate priors within the Diffusion Model to accomplish the FSS task.

Recently, inspired by the in-context learning ability of large language models, Few-shot Semantic Segmentation has further evolved into the In-context Segmentation [29, 30] task (see Section 2). In-context Segmentation requires the model to have in-context learning ability for few-shot samples, posing new challenges to model's generalization capabilities. Consequently, it's now recognized as a crucial component in the evaluation process for generalist segmentation models. Therefore, **the second motivation** of our work is to lay the groundwork for the development of diffusion-based generalist segmentation models.

As a foundational work of Diffusion-based methods in the FSS field, we strive to achieve optimal performance with a simple and efficient design, while maximally preserving the generative framework of the Latent Diffusion Model. This minimal disruption to the original UNet structure allows us to better make use of pre-trained priors. We embark on a systematic exploration around the following four questions: 1) How to implement the interaction between the query image and the support image? 2) How to effectively inject information from the support mask? 3) What is a reasonable form of supervision from the query mask? 4) How to design effective generation process to transfer the pre-trained diffusion models to mask prediction task? Based on our observations, we ultimately establish the DiffewS framework and validate it in multiple settings, demonstrating the effectiveness of our method. Our main contributions include:

- We systematically study four crucial elements of applying the Diffusion Model to Few-shot Semantic Segmentation. For each of these aspects, we propose several reasonable solutions and validate them through comprehensive experiments.

- Building upon our observations, we establish the DiffewS framework, which maximally retains the generative framework and effectively utilizes the pre-training prior. Notably, we introduce the first diffusion-based model dedicated to Few-shot Semantic Segmentation, setting the groundwork for a diffusion-based generalist segmentation model.

- We validate the effectiveness of the DiffewS framework under several experimental settings, demonstrating that our method not only achieves a performance comparable with the state-of-the-art (SOTA) model in a strict Few-shot Semantic Segmentation setting, but also significantly outperforms the current SOTA model in an 'in-context learning' setting,

## 2   Related Work

**Diffusion models** have shown impressive performance on visual generation tasks such as text-based image generation [1, 2], image editing [3, 4], and video generation [5–7]. Current research on leveraging Diffusion models to enhance visual perception tasks mainly focuses on two directions: one is the direct use of diffusion models to generate images, aiming to address the issue of insufficient data, such as instance segmentation [31–33], semantic segmentation [34], few-shot segmentation [35] and so on. Another direction is to transfer features from Diffusion models to other visual tasks, which aligns with the research direction of this paper.

ODISE [16] uses frozen diffusion models for panoptic segmentation of any category in the wild. DVP [12], DatasetDM [17], GenPercept [36], Geowizard [37] adapt LDM to multiple tasks such as depth estimation, semantic segmentation, and surface normal. Marigold [13] fine-tunes diffusion models on synthetic data for affine-invariant monocular depth estimation and achieves impressive performance. Different from the above methods, we focus on using diffusion models to model the visual correlations of multiple reference images and a target image for few-shot segmentation. The most related work to this paper is a concurrent study [38], which focuses on utilizing diffusion models for in-context segmentation. However, it disrupts the original U-Net structure and the priors of the diffusion model to some extent. In contrast, our work offers a more comprehensive and systematic analysis of applying diffusion models to Few-Shot Semantic Segmentation tasks.

**Few-shot semantic segmentation** [39, 40] aims to segment target objects in an input image given a few annotated support images. Traditional FSS methods either explore prototype learning [41–43] of support images to predict query images' masks or use pixel-level information [44, 45, 18] to exploit the support information. For example, some works [29, 30, 46] demonstrate powerful generalization ability by unifying various segmentation tasks in an in-context learning framework. SegGPT [30] can exactly segment any semantic conception by using one or a few support images, which motivates us to explore the potential of the diffusion model for the FSS task under the in-context setting [30].

## 3 Preliminary

We first review the Latent Diffusion Model [2] used in our paper. It consists of an auto-encoder (VAE) and a UNet. The auto-encoder facilitates a two-way transformation between the RGB image $\mathbf{I} \in \mathbb{R}^{H \times W \times 3}$ and the latent space $\mathbf{z} \in \mathbb{R}^{h \times w \times c}$. Both the forward and backward processes of diffusion are carried out in the latent space, and we denote the noisy latent code at time $t$ as $\mathbf{z}^{(t)} = \sqrt{\bar{\alpha}_t}\mathbf{z} + \sqrt{1 - \bar{\alpha}_t}\epsilon$, where $\bar{\alpha}_t = \prod_{s=1}^{t}(1 - \beta_s)$ is the noise schedule. $\beta_s$ is the variance sampled from a variance schedule $\beta_t \in (0, 1)_{t=1}^{T}$. The UNet can be considered as a series of equally weighted denoiser $\epsilon_\theta(\mathbf{z}^{(t)}, t)$. The training objective $\mathcal{L}$ can be simplified as:

$$\mathcal{L} = \mathbb{E}_{\mathbf{z}, \epsilon \sim \mathcal{N}(0,1), t \in \mathcal{U}(T)} \left[ \left\| \epsilon - \epsilon_\theta \left( \mathbf{z}^{(t)}, t \right) \right\|_2^2 \right] \tag{1}$$

Furthermore, to simplify comprehension and narration, we can reparametrize the output of UNet $\epsilon_\theta$ as the form of v-prediciton $v_\theta$. The training objective can be further elaborated as:

$$\mathcal{L} = \mathbb{E}_{\mathbf{z}, \epsilon \sim \mathcal{N}(0,1), t \in \mathcal{U}(T)} \left[ \left\| \mathbf{z} - v_\theta \left( \mathbf{z}^{(t)}, t \right) \right\|_2^2 \right] \tag{2}$$

This implies that the goal of every training round is to denoise $\mathbf{z}^{(t)}$ to $\mathbf{z}$ for any time step $t$.

Secondly, we present our task definition, using one-shot segmentation as an illustration. Given a data triplet $(\mathbf{I}_s, \mathbf{M}_s, \mathbf{I}_q)$, here $\mathbf{I}_s$ and $\mathbf{I}_q$ denote the support image and query image respectively, both sharing an overlapping category $c$. $\mathbf{M}_s$ is the mask of category $c$ in the support image. Our task is to predict the mask corresponding to category $c$ in $\mathbf{I}_q$. In the strict one-shot segmentation setting, the category sets of the training set and the test set are disjoint.

Our objective is to fully utilize the priors in the Latent Diffusion Model and equip it with Few-shot Semantic Segmentation capabilities. This leads us to reuse the original VAE to convert $\mathbf{I}_s$, $\mathbf{I}_q$ and $\mathbf{M}_q$ into latent variables $\mathbf{z}_s$, $\mathbf{z}_q$ and $\mathbf{z}_{mq}$. Thus, our task is further simplified to explore how to improve the structure of UNet to $v_\theta^*$ so that it can accept $\mathbf{z}_s$, $\mathbf{z}_q$ and $\mathbf{M}_s$ as inputs, and use $\mathbf{z}_{mq}$ as supervision.

This supervised approach in the latent space has been certified effective in tasks such as depth estimation [13] and semantic segmentation[14]. Concretely, our training objective $\mathcal{L}_{\mathcal{FSS}}$ is transformed into:

$$\mathcal{L}_{\mathcal{FSS}} = \mathbb{E}_{(\mathbf{z}_s, \mathbf{z}_q, \mathbf{M}_s, \mathbf{z}_{mq}) \sim \mathcal{D}} \left[ \left\| \mathbf{z}_{mq} - v_\theta^* \left( \mathbf{z}_s, \mathbf{z}_q, \mathbf{M}_s \right) \right\|_2^2 \right] \tag{3}$$

where $\mathcal{D}$ represents the constructed training dataset. In addition, we omitted the input of time $t$. Our early experiments revealed that performing multiple steps of noise addition and denoising during training did not bring performance improvement.

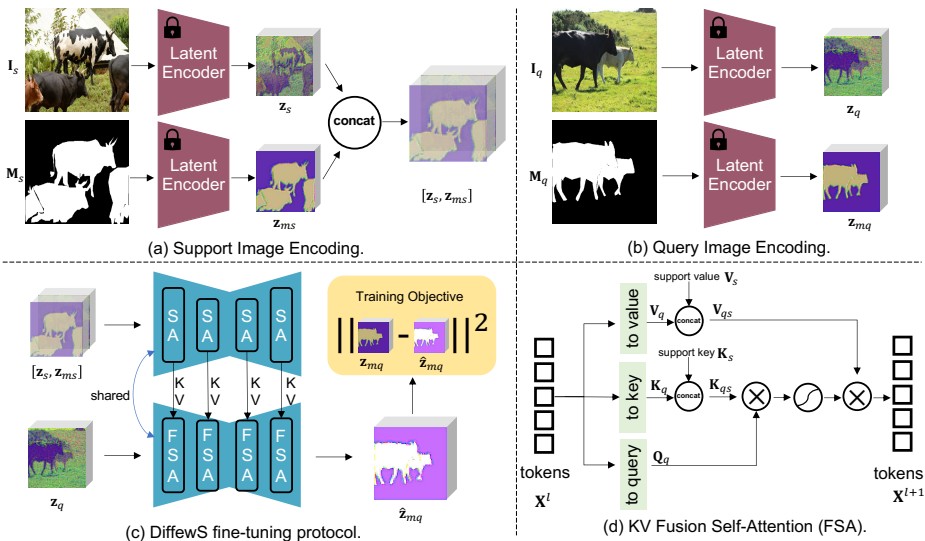

(a) Support Image Encoding.

(b) Query Image Encoding.

(c) DiffewS fine-tuning protocol.

(d) KV Fusion Self-Attention (FSA).

**Figure 1** – Overview of the DiffewS framework. (a)(b) display that query image $\mathbf{I}_q$, query mask $\mathbf{M}_q$, support image $\mathbf{I}_s$ and support mask $\mathbf{M}_s$ are all encoded by VAE into latent variables $\mathbf{z}_q$, $\mathbf{z}_{mq}$, $\mathbf{z}_s$, $\mathbf{z}_{ms}$, respectively, where $\mathbf{z}_q$ and $\mathbf{z}_{mq}$ are concatenated to input into UNet. (c) demonstrates the DiffewS fintuning protocol (d) elucidates the detailed implementation of FSA, acquiring information from support images by concatenating the query and key features.

## 4 Method

Our investigation into model design primarily adheres to two criteria: 1. Strive for the design to be as simple and efficient as possible, while optimizing performance in Few-shot Semantic Segmentation. 2. Maximize the preservation of the Latent Diffusion Model's generative schema, minimizing alteration to the original UNet structure, so as to better utilize the pre-training prior.

Specifically, four key issues need to be addressed: 1) How to facilitate interaction between the query image and support image? 2) How to effectively incorporate information from the support mask? 3) What form of supervision from the query mask would be most reasonable? 4) How to design an effective generation process to transfer the pre-trained diffusion models to mask prediction task? In this section, we discuss the four issues mentioned above in detail. We engage in fair comparison tests and analysis on several feasible strategies. Drawing on our observations, we eventually settle on our framework, DiffewS (see Figure 1).

### 4.1 Interaction between query and support images

We first decompose the block of the l-th layer in UNet into three components: a self-attention layer $\mathrm{SelfAttn}$, a cross-attention layer $\mathrm{CrossAttn}$, and a feedforward layer $\mathrm{FFN}$. Given the feature map $\mathbf{X}^l$ of the l-th image and the textual input $\mathbf{t}$ (which is an empty character in our task), we obtain:

$$\mathbf{X}^{l+1} = \mathrm{FFN}\left(\mathrm{CrossAttn}\left(\mathrm{SelfAttn}\left(\mathbf{X}^l\right), \mathrm{CLIP}_{text}(\mathbf{t})\right)\right), \tag{4}$$

where $\mathrm{CLIP}_{text}$ represents CLIP text encoder, and we have skipped over skip-connection in the formula.

Before considering the incorporation of the support mask, two straightforward and intuitive methods can be leveraged to facilitate interaction between the query image and support image. One approach entails interaction within the self-attention module, while the other involves interaction within the cross-attention module.

**KV Fusion Self-Attention.** We first propose a KV fusion method in self-attention layer to achieve interaction between query image and support image. For the input image feature $\mathbf{X}$, the standard self-attention layer first maps it to query $\mathbf{Q}$, key $\mathbf{K}$ and value $\mathbf{V}$ with a linear projection layer. . Therefore, $\mathrm{SelfAttn}(\mathbf{X})$ can be further represented as:

$$\mathbf{X}^* = \mathrm{SelfAttn}(\mathbf{X}) = \mathrm{Attention}(\mathbf{Q}, \mathbf{K}, \mathbf{V}) = \mathrm{Softmax}(\frac{\mathbf{Q}\mathbf{K}^T}{\sqrt{d}})\mathbf{V} \tag{5}$$

where $d$ is the dimension of query and key, while $\mathbf{X}^*$ is the feature updated by self-attention. Back to our task, we can also map the features of the support image and query image $\mathbf{X}_s$ and $\mathbf{X}_q$ to $\mathbf{Q}_s$, $\mathbf{K}_s$, $\mathbf{V}_s$ and $\mathbf{Q}_q$, $\mathbf{K}_q$, $\mathbf{V}_q$ through the linear projection layer. We hope that the features of the query image can effectively utilize the information of the support image, so we need to let $\mathbf{Q}_q$ access $\mathbf{K}_s$ and $\mathbf{V}_s$. To achieve this, we can concatenate $\mathbf{K}_q$ and $\mathbf{K}_s$ to form $\mathbf{K}_{qs} = [\mathbf{K}_q, \mathbf{K}_s]$. Similarly, we can get $\mathbf{V}_{qs} = [\mathbf{V}_q, \mathbf{V}_s]$. Finally, our KV Fusion Self-Attention layer can be represented as:

$$\mathbf{X}_q^* = \text{FusionAttn}(\mathbf{X}_q, \mathbf{X}_s) = \text{Attention}(\mathbf{Q}_q, \mathbf{K}_{qs}, \mathbf{V}_{qs}) \tag{6}$$

Since we only replaced $\mathbf{K}$ and $\mathbf{V}$, we can fully reuse the weights of the original self-attention.

**Tokenized Interaction Cross-Attention** The second alternative is to inject information originating from the support image via cross-attention. This strategy has been widely used in Customized Text-to-Image Generation [47–49]. In particular, the initial cross-attention is employed to introduce the text information, encoded using CLIP text encoder. We can encode the support image into a series of tokens using the CLIP image encoder and utilize it as the cross-attention input. At this point, the process can be represented as:

$$\mathbf{X_q}^* = \text{CrossAttn}(\mathbf{X_q}, \text{Flatten}(\text{CLIP}_{img}(\mathbf{I}_s))) \tag{7}$$

where $\text{Flatten}$ means flattening the token sequence after image encoding. $\text{CLIP}_{img}$ represents the CLIP image encoder corresponding to the CLIP text encoder used in the original UNet.

### 4.2 Injection of support mask information

Building upon the Self-attention kv fusion approach, we investigate methodologies for incorporating support mask information. We categorize the injection methods into four types:

a. **Concatenation** The support mask $\mathbf{M}_s$ can be converted into an RGB image, then directly encoded into a latent variable $\mathbf{z}_{ms}$ using VAE, which is then concatenated with $\mathbf{z}_s$ in the channel dimension. Due to the resulting mismatch in dimensionality from the concatenation, we adopt the approach of Marigold [13], where the first layer weight tensor is duplicated and its values are halved.

b. **Multiplication** We can directly multiply $\mathbf{M}_s$ on the image $\mathbf{I}_s$ to form the image $\mathbf{I}_s^* = \mathbf{I}_s \cdot \mathbf{M}_s$, and finally encode $\mathbf{I}_s^*$ into a latent variable $\mathbf{z}_s^*$ using VAE as the input of UNet.

c. **Attention Mask** $\mathbf{M}_s$ can serve as an attention mask to control self-attention so that only $\mathbf{K}_s$ in the masked region can be accessed by $\mathbf{Q}_q$. Since the feature map sizes of different layers are different, we need to resize $\mathbf{M}_s$ to fit the dimensions of each layer.

d. **Addition** Alternatively, $\mathbf{M}_s$ can be directly added to the image $\mathbf{I}_s$, generating the image $\mathbf{I}_s^* = 0.5\mathbf{I}_s + 0.5\mathbf{M}_s$. Following that, $\mathbf{I}_s^*$ is encoded into a latent variable $\mathbf{z}_s^*$ using VAE, which is then used as the UNet input.

For cross-attention tokenized interaction, information of the support mask can also be injected in the same four ways. There are just some slight differences in the implementation details (see the Appendix A.3).

We carry out a comparison of two interaction methods (Section 4.1) paired with four injection methods (Section 4.2); these eight combinations are then verified experimentally, and the results are presented in Figure 2. Overall, we observe that KV Fusion Self-Attention(FSA) outperforms Tokenized Interaction Cross-Attention(TCA). We attribute this mainly to the preservation and flexible utilization of information from the support image by FSA. Conversely, TCA, which only compresses support image to tokens via the CLIP image encoder, leads to some information loss. Notably, within the FSA, the Concatenation method surpassed the other three. It offered a more free-form handling of RGB images and MASK information via subsequent learnable convolutional layers, compared

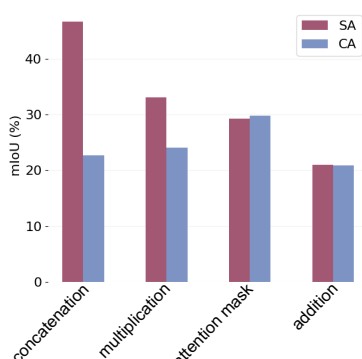

**Figure 2** – Exploring the Interaction and Injection Methods

to other hard injection methods. In the case of TCA, the Attention Mask method seems more apt as other operations are actually constrained by the CLIP image encoder. The CLIP image encoder itself is not good at dealing with mask information. Of course, we believe that there is still room for further exploration here, referring to FGVP [50].

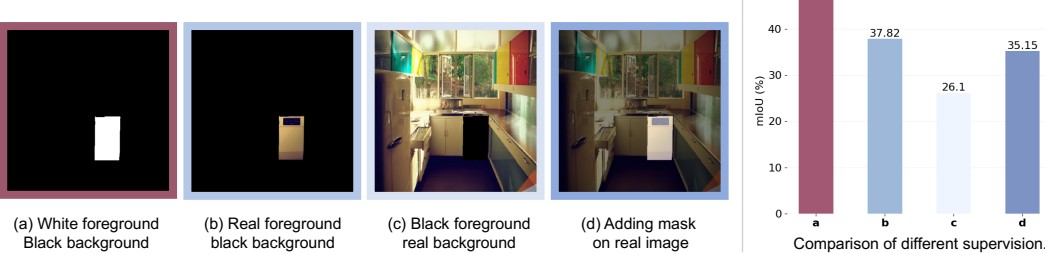

**Figure 3** – Illustrations and comparisons of different forms of supervision from query mask.

### 4.3 Supervision from query mask

In Section 3, we mentioned that we encode the query mask $\mathbf{M}_q$ into a latent variable $\mathbf{z}_{mq}$, and directly supervise in the latent space. However, $\mathbf{M}_q \in [0, 1]^{H \times W}$ is a two-dimensional mask, while the input of VAE needs to be an RGB image. Consequently, conversion of $\mathbf{M}_q$ into an RGB image becomes necessary, but it's unclear which form of conversion would yield optimal results as no research has delved into this as yet. A reasonable conversion method should satisfy the following two conditions:1. It is easier for UNet to learn 2. It is more convenient to get the final segmentation result through post-processing. In this section, we explore the following four forms of conversion.

**a. White foreground + black background** Visualizing the segmentation annotation with a white mask and black background is a common way in the academic community. Specifically, we only need to replicate $\mathbf{M}_q$ three thrice along the channel dimension to form the corresponding RGB image denoted by the mask. We employed this conversion approach as a default in Section 4.2.

**b. Real foreground + black background** Considering LDM's original pre-training on real images, forcing the model to output purely black-and-white images that do not fit within real-image distribution might amplify the model's learning difficulty. Therefore, we also attempted to retain the real pixels of the foreground, while setting the background to black

**c. Black foreground + real background** Following the same logic, we also try preserving the pixels of the real background but render the foreground pixel black.

**d. Adding mask on real image** We also consider overlaying $\mathbf{M}_q$ on the real image to form the mask on the real image, which is the Addition method mentioned in Section 4.2. This approach makes the output space of UNet closer to the distribution of real images, but it requires more complex post-processing to get the final segmentation results. That is, we need to subtract the original image from the model output to get the final segmentation result.

As shown in Figure 3, we assess the performance of the four forms of supervision, among which (a) method achieved the best performance in all experiments. Although (b) (c) (d) methods being closer to the real image distribution, the performance is lower. On the one hand, it is difficult to obtain the mask through simple post-processing, and on the other hand, it may increase the learning difficulty because the model needs to retain the ability to generate the original image. In conclusion, our results demonstrate that UNetUNet can effortlessly learn to output in forms such as 'white foreground + black background'. Therefore, we eventually chose this approach for Diffews.

### 4.4 Exploration of generation process

In this section, we further discuss how to design an effective generation process to transfer the pre-trained diffusion models to mask prediction tasks. Inspired by the success of transferring pre-trained diffusion models to depth estimation task [13, 51], we explore three different mask generation processes. The illustration of different mask generation processes is shown in Figure 4.

- **Multi-step noise-to-mask generation (MN2M)** MN2M follows the denoise pipeline of original diffusion models. The training and inference schemes of MN2M are similar to Marigold [13]. Figure 4(b1) shows the illustration of inference process. The image latent $z_q$

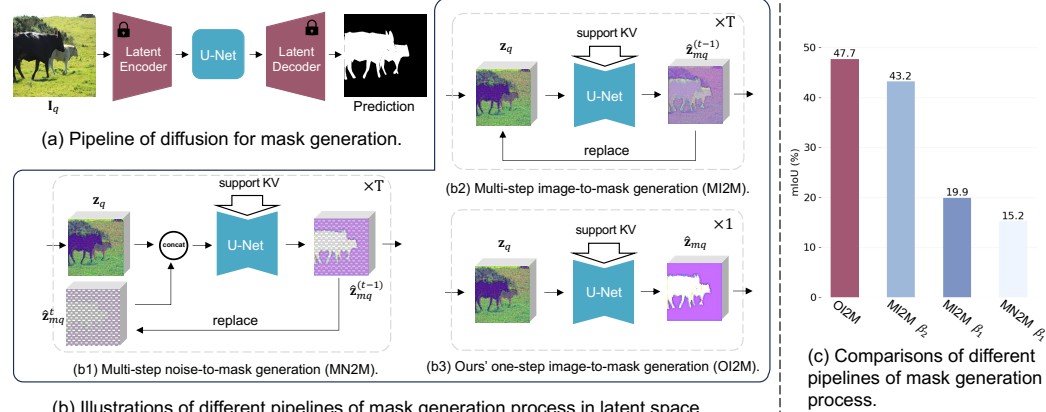

(a) Pipeline of diffusion for mask generation.

(b1) Multi-step noise-to-mask generation (MN2M).

(b2) Multi-step image-to-mask generation (MI2M).

(b3) Ours' one-step image-to-mask generation (OI2M).

(b) Illustrations of different pipelines of mask generation process in latent space.

(c) Comparisons of different pipelines of mask generation process.

**Figure 4** – Illustrations and comparisons of different mask generation processes.

concatenates with the mask latent $\hat{\mathbf{z}}_{mq}^{(t)}$. The UNet takes it as input and predicts the new mask latent $\hat{\mathbf{z}}_{mq}^{(t-1)}$. After T steps, the final mask latent $\hat{\mathbf{z}}_{mq}^{(0)}$ is decoded into mask prediction. The mask latent $\hat{\mathbf{z}}_{mq}^{(T)}$ is initialized as random noise. We also use the annealed multi-resolution noise and test-time ensemble tricks [13] proposed in Marigold.

- **Multi-step image-to-mask generation (MI2M)** MI2M formulates the diffusion denoising process as a deterministic multi-step conversion process from image to prediction, similar to DMP [51]. Figure 4(b2) shows the illustration of inference process. The mask latent $\hat{\mathbf{z}}_{mq}^{(T)}$ is initialized as image latent $\mathbf{z}_q$. Then similar to MN2M, the UNet takes $\hat{\mathbf{z}}_{mq}^{(t)}$ as input and predicts $\hat{\mathbf{z}}_{mq}^{(t-1)}$. After T steps, the final mask latent $\hat{\mathbf{z}}_{mq}^{(0)}$ is decoded into mask prediction.

- **One-step image-to-mask generation (OI2M)** OI2M further transforms MI2M's multi-step prediction into a one-step prediction, *i.e.*, UNet takes $\mathbf{z}_q$ as input and outputs the prediction $\hat{\mathbf{z}}_{mq}$ directly.

We explore the mask generation pipeline starting from MN2M. As shown in Figure 4(c), MN2N achieves 15.2% mIoU. Then, we change MN2M into MI2M keeping same variance $\beta^1 = (0.00085, 0.012)$, respectively representing the initial and final values of $\beta$ in the DDIM scheduler. The performance has improved by 4.7% mIoU. However, despite the improvement, both methods exhibit suboptimal performance. We hypothesize that this is because adding a very small noise or image to the binary mask during the training process and then predicting it does not lead to a challenging task compared with diffusion pre-training.

We hypothesize that the suboptimal performance is due to the minimal noise or image added to the binary mask during training, which results in an insufficiently challenging task compared to diffusion pre-training. The binary mask is inherently simpler than natural images, and even after adding noise, the latent mask can still easily distinguish between the foreground and background. This simplicity causes significant information leakage during UNet training, ultimately leading to poor performance.

To verify this hypothesis, we increase the variance of MI2M from $\beta^1 = (0.00085, 0.012)$ to $\beta^2 = (0.0272, 0.384)$. The performance has significantly improved by 23.3% mIoU. To fully increase the challenge of training, we convert MI2M into OI2M, which does not introduce any ground-truth information into the input of the UNet during training. Additionally, OI2M reduces the number of iterations to one, significantly boosting the network's predictive efficiency. As shown in Figure 4(c), OI2M achieves the best performance, making it the preferred choice for the mask generation pipeline.

### 4.5   1-shot to N-shot

So far, we have primarily explored the training and inference processes specifically designed for 1-shot scenarios. A natural question arises: can this framework be extended to n-shot settings? To address this, we first present the simplest and most straightforward method for adaptation, which requires only minor modifications during the inference phase to accommodate n-shot tasks.

In the Section 4.1, we introduced how to inject the information of the support image into the features of the query image using the KV Fusion Self-Attention method. In inference, our support set $S$ may contain more than one image, $S = \{I_{s1}, I_{s2}, ..., I_{sn}\}$. We encode each image into the features $\mathbf{X}_{si}$. Correspondingly, after mapping, we can obtain a series of $\mathbf{Q}_{si}, \mathbf{K}_{si}, \mathbf{V}_{si}$ and $\mathbf{Q}_{qi}, \mathbf{K}_{qi}, \mathbf{V}_{qi}$. We can concatenate $\mathbf{K}_{qi}$ and $\mathbf{K}_{si}$ to form $\mathbf{K}_{qs} = [\mathbf{K}_{qi}, \mathbf{K}_{s1}, \mathbf{K}_{s2}, ..., \mathbf{K}_{sn}]$, and similarly we can obtain $\mathbf{V}_{qs} = [\mathbf{V}_{qi}, \mathbf{V}_{s1}, \mathbf{V}_{s2}, ..., \mathbf{V}_{sn}]$. Finally, our kv fusion self attention layer can be represented as:

$$\mathbf{X}_q^* = KVFusionAttn(\mathbf{X}_q, \mathbf{X}_s) = Attention(\mathbf{Q}_q, \mathbf{K}_{qs}, \mathbf{V}_{qs}) \tag{8}$$

While the aforementioned solutions enable N-shot inference, their performance does not match that of state-of-the-art (SOTA) models. This discrepancy primarily arises because the model receives only a single support image during the training phase, which leads to inconsistencies when transitioning to the inference phase with 5-shot or 10-shot configurations.

To address this issue, we explore improvements from both the inference and training perspectives. From the perspective of inference, transitioning from 1-shot to N-shot involves concatenating the keys and values of additional support samples, which significantly increases the number of keys and values processed during inference. To address this, we implement random sampling of the keys and values from the support samples during inference, ensuring that their quantity matches that of the training phase (see Table 6). Another more straightforward idea is to introduce multiple support samples during the training phase. In this way, the model can learn how to utilize multiple support images during training. we randomly select 1 to N support samples as input using KV Fusion in Equation (8) during a single training iteration (see Table 7).

Our experiments demonstrate that improvements during the training phase are more effective than those during the inference phase. Therefore, we include the results of the model with training phase improvements in Table 2.

## 5 Experiment

**Datasets** We test our method in two settings: 1. Strict few shot setting: Following the few-shot setting on COCO-20$^i$ [52], we organize 80 classes from COCO2014 [53] into 4 folds. Each trial consists of 60 classes allocated for training and 20 classes designated for testing. For evaluation, we randomly sample 1000 reference-target pairs in each fold with the same seed used in HSNet [18]. 2. In-context setting: Following the setting in SegGPT [30], COCO, ADE [54], and PASCAL VOC [55] serve as the training set. In-domain testing is conducted on COCO-20$^i$ and PASCAL-5$^i$ [39] to evaluate our model. In line with Matcher [22], LVIS-92$^i$ function as the out-of-domain test set.

**Implementation details** We initialize our model with Stable Diffusion 2.1 [2]. The Adam optimizer is used with a weight decay set at 0.01 and a learning rate of 1e-5, coupled with a linear schedule. In terms of data augmentation, our methodology only involves resizing the input image directly to 512x512. No additional data augmentation occurs. Under the strict few-shot setting, the model undergoes training on four V100 GPUs. With the gradient accumulation set at 4, the total batch size comes to 16. Training carries out for 10,000 iterations, typically requiring six hours. For in-context setting, since the training set is larger, we keep other hyperparameters consistent with the strict few-shot setting, and adjust the total training iterations to 30000 iterations. Lastly, our ablation experiments are validated on Fold0 of COCO-20$^i$ [52]. The training took place on a single 4090 GPU, with a gradient accumulation set at 4, which brought the total batch size to 4. The training, which consisted of 10,000 iterations, took roughly 11 hours.

### 5.1 In-context setting

We first compare DiffewS with other generalist models such as Painter [29], SegGPT [29], PerSAM-F[58], and Matcher [22] as well as specialist models like HSNet [18], VAT [56], FPTrans [57]. Regarding the specialist models, we directly refer to the results presented within the SegGPT [30] and Matcher [22] research papers. These specialist models are also trained on the test categories from COCO [53] and PASCAL VOC [55]. We employ COCO-20$^i$ [52] and PASCAL-5$^i$ [39] to validate the in-domain performance of DiffewS. Remarkably, on COCO, DiffewS achieves a 1-shot score of 71.3, considerably exceeding the generalist model SegGPT (+15.2) and specialist model FPTrans (+14.8), both trained with in-domain data. DiffewS furthermore significantly outperforms

**Table 1** – Results of few-shot semantic segmentation on COCO-20$^i$, PASCAL-5$^i$, and LVIS-92$^i$, under in-context setting.

| Methods | Venue | COCO-20$^i$ | | PASCAL-5$^i$ | | LVIS-92$^i$ | |
|---|---|---|---|---|---|---|---|
| | | one-shot | few-shot | one-shot | few-shot | one-shot | few-shot |
| HSNet [18] | ICCV'21 | 41.7 | 50.7 | 68.7 | 73.8 | 17.4 | 22.9 |
| VAT [56] | ECCV'22 | 42.9 | 49.4 | 72.4 | 76.3 | 18.5 | 22.7 |
| FPTrans [57] | NeurIPS'22 | 56.5 | 65.5 | 77.7 | 83.2 | - | - |
| Painter [29] | CVPR'23 | 32.8 | 32.6 | 64.5 | 64.6 | 10.5 | 10.9 |
| SegGPT [30] | ICCV'23 | 56.1 | 67.9 | 83.2 | 89.8 | 18.6 | 25.4 |
| PerSAM [58] | ICLR'24 | 23.0 | - | - | - | 15.6 | - |
| PerSAM-F [58] | | 23.5 | - | - | - | 18.4 | - |
| Matcher [22] | ICLR'24 | 52.7 | 60.7 | 67.9 | 75.6 | 33.0 | 40.0 |
| DiffewS | this work | 71.3 | 72.2 | 88.3 | 87.8 | 31.4 | 35.4 |

**Table 2** – Results of strict few-shot semantic segmentation on COCO-20$^i$. DiffewS-n represents using training time improvements for N-shot.

| Methods | Venue | 1-shot | | | | | 5-shot | | | | |
|---|---|---|---|---|---|---|---|---|---|---|---|
| | | $20^0$ | $20^1$ | $20^2$ | $20^3$ | mean | $20^0$ | $20^1$ | $20^2$ | $20^3$ | mean |
| HSNet [18] | ICCV'21 | 37.2 | 44.1 | 42.4 | 41.3 | 41.2 | 45.9 | 53.0 | 51.8 | 47.1 | 49.5 |
| CyCTR [59] | NeurIPS'21 | 38.9 | 43.0 | 39.6 | 39.8 | 40.3 | 41.1 | 48.9 | 45.2 | 47.0 | 45.6 |
| VAT [56] | ECCV'22 | 39.0 | 43.8 | 42.6 | 39.7 | 41.3 | 44.1 | 51.1 | 50.2 | 46.1 | 47.9 |
| BAM [60] | CVPR'22 | 43.4 | 50.6 | 47.5 | 43.4 | 46.2 | 49.3 | 54.2 | 51.6 | 49.6 | 51.2 |
| DCAMA [19] | ECCV'22 | 49.5 | 52.7 | 52.8 | 48.7 | 50.9 | 55.4 | 60.3 | 59.9 | 57.5 | 58.3 |
| HDMNet [20] | CVPR'23 | 43.8 | 55.3 | 51.6 | 49.4 | 50.0 | 50.6 | 61.6 | 55.7 | 56.0 | 56.0 |
| DiffewS | this work | 47.7 | 56.4 | 51.9 | 48.7 | 51.2 | 52.0 | 63.0 | 54.5 | 54.3 | 56.0 |
| DiffewS-n | | 47.1 | 56.6 | 53.8 | 48.3 | 52.2 | 57.3 | 66.5 | 60.3 | 58.8 | 60.7 |

SAM-based models PerSAM-F (+47.8) and Matcher (+18.6). On PASCAL-5$^i$, DiffewS records 88.3 in 1-shot, clearly surpassing SegGPT (+5.1) and Matcher (+20.4). These results evidence that DiffewS effectively utilizes the prior of Stable Diffusion, unlocking the full potential of Stable Diffusion in segmentation. Furthermore, out-of-domain examination on LVIS-92$^i$ [22] underpins the generalization ability of DiffewS. In this setting, DiffewS registers 31.4 in 1-shot and 35.4 in 5-shot, markedly outperforming other generalist models, aside from Matcher. It is worth mentioning that Matcher simultaneously utilizes two Foundation models (SAM [21] and DINO V2 [23] ), and SAM itself is pre-trained on an exhaustive, finely annotated segmentation dataset. On the other hand, DiffewS undergoes fine-tuning on a relatively smaller quantity of segmentation data for limited iterations, still delivering performance that rivals Matcher. This indicates that using the paradigm of DiffewS, there is potential to achieve significant breakthroughs in the segmentation field if further trained on larger-scale segmentation data. It should be noted that the improvement of DiffewS in 5-shot is not significant, with a 4.0 distinct improvement only on LVIS-92$^i$. This might be due to the presence of many small objects in the support images of LVIS, so increasing the number of support images can alleviate this problem. Conversely, the DiffewS 5-shot performance on PASCAL-5$^i$ [39] is slightly deficient compared to the 1-shot. This could be ascribed to the presence of relatively larger and more simplistic objects within PASCAL VOC's support images, inputting more images might interfere with the original architecture of the model. In this case, we do not apply the improvement strategies discussed in Section 4.5, therefore, the relatively weaker performance in the 5-shot scenario is reasonable.

## 5.2 Strict few-shot setting

We also undertake validation of DiffewS under the standard few-shot setting, comparing it with other specialist models such as HSNet [18], CyCTR [59], VAT [56], BAM [60], HDMNet [20], and DCAMA [19]. For the one-shot setting, the average performance of DiffewS across all four folds attains 51.2, surpassing the current state-of-the-art (SOTA) model DCAMA, scoring 50.9 mIoU. Worth mentioning is that DCAMA relies on a highly complex additional block, whereas DiffewS entirely utilizes the generative framework of UNet. In terms of the efficiency of convergence, DiffewS necessitates just a 30000-iteration training, in contrast to both DCAMA and HSNet which require training spanning hundreds of epochs, typically costing several days. This demonstrates the successful employment of Stable Diffusion priors by DiffewS, thereby securing impressive performance without requiring extended periods of fine-tuning. In the five-shot setting, the average performance across four

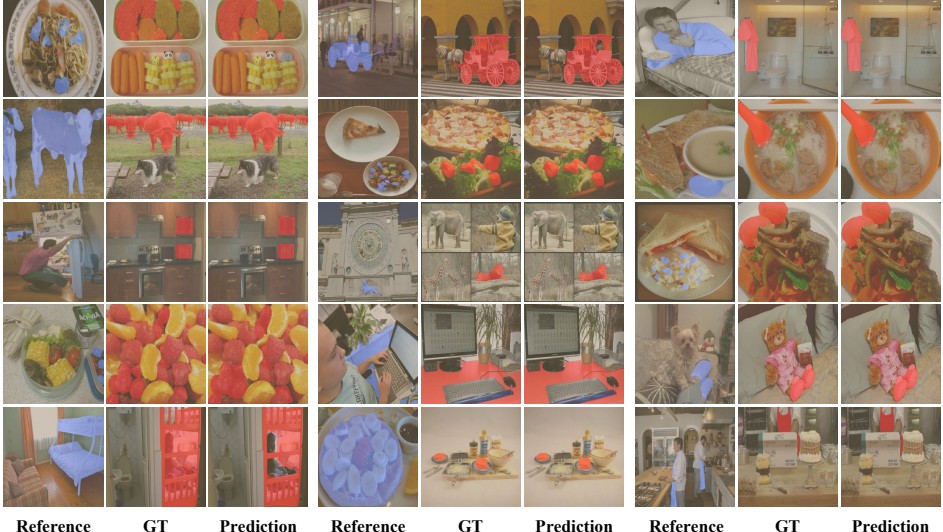

| Reference | GT | Prediction | Reference | GT | Prediction | Reference | GT | Prediction |

**Figure 5** – Qualitative results of one-shot semantic segmentation on LVIS-92$^i$. The blue color denotes the support mask while the red represents the query mask.

folds reaches 56.0, higher than all other models aside from DCAMA. Currently, DiffewS primarily focuses on the 1-shot situation lacking specific optimizations for the 5-shot scenario in its training and inference systems. This explains why DiffewS is at present marginally inferior to DCAMA. Furthermore, when employing our proposed training improvement strategy, DiffewS-n outperforms other models in both the 1-shot and 5-shot settings.

### 5.3 Visualization

As shown in Figure 5, DiffewS effectively segments categories not in the training set, such as slippers and aprons. It also accurately segments objects of different styles and smaller items, demonstrating strong generalization capabilities. In some cases, DiffewS even achieves more accurate results than GT.

In addition, DiffewS demonstrates impressive results in various cross-style segmentation tasks and small object segmentation cases (see Figure 6). We hypothesize that DiffewS's exceptional generalization ability stems from its extensive utilization of prior knowledge from diffusion models. However, DiffewS also struggles with certain challenging cases, we also present several failure cases in Figure 7 and categorize the reasons for these failures.

## 6  Conclusion

In this work, we have presented DiffewS, a simple and efficient framework for few-shot semantic segmentation. By directly generating the target mask, DiffewS is capable of retaining the original latent diffusion models' generative framework and effectively utilizing the visual prior of pre-trained diffusion models. By introducing several designs about multi-image interaction, information injection, and supervision signals, DiffewS outperforms SOTA models in the in-context learning setting, and reaches comparable performance to specialist models in the strict few-shot setting.

**Limitation** and more **Discussions** are provided in Appendix A.1.

## Acknowledgement

This work is partially supported by the National Key R&D Program of China(NO.2022ZD0160101) and the National Natural Science Foundation of China (No. 62206244).

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

# A   Appendix

### A.1   Discussion

**Broader Impacts** We do not foresee any obvious undesirable ethical or social impacts now.

**Limitations** Our method, as the first diffusion-based FSS model, proposes a simple and intuitive design, which maximizes the retention of the generative framework of LDM. There is still a lot of room for improvement in performance (especially in the n-shot setting), including more sophisticated model design and more optimized training strategies. We hope that our method can serve as a diffusion-based FSS baseline to inspire more researchers to invest in this field.

On the other hand, we believe that our method is not limited to FSS. Our framework has the potential to unify few-shot segmentation and open vocabulary segmentation by leveraging prompts from different modalities, as some work [61–63] has already proven the possibility.

## A.2 More details on generation process

In the above section we have discussed the generation process of DiffewS. In addition to the final choice of OI2M, we also tried MN2M and MI2M. Here we detail the training objectives of these three generation processes.

**OI2M** We directly input the image and let the UNet output the mask. This process can be described as:

$$\mathcal{L}_{\textbf{OI2M}} = \mathbb{E}_{(\mathbf{z}_s, \mathbf{z}_q, \mathbf{z}_{ms}, \mathbf{z}_{mq}) \sim \mathcal{D}} \left[ \left\| \mathbf{z}_{mq} - v_\theta^* \left( \mathbf{z}_s, \mathbf{z}_q, \mathbf{z}_{ms} \right) \right\|_2^2 \right] \tag{9}$$

**MN2M** We add noise to query mask $\mathbf{z}_{mq}$, $\mathbf{z}_{mq}^{(t)} = \sqrt{\bar{\alpha}_t} \mathbf{z}_{mq} + \sqrt{1 - \bar{\alpha}_t} \epsilon$, and during inference we use $\mathbf{z}_{mq}^{(0)}$ as the mask prediction. The supervised form is as follows:

$$\mathcal{L}_{\textbf{MN2M}} = \mathbb{E}_{(\mathbf{z}_s, \mathbf{z}_q, \mathbf{z}_{ms}, \mathbf{z}_{mq}) \sim \mathcal{D}, \epsilon \sim \mathcal{N}(0,1), t \in \mathcal{U}(T)} \left[ \left\| \mathbf{z}_{mq} - v_\theta^* \left( \mathbf{z}_{mq}^{(t)}, \mathbf{z}_s, \mathbf{z}_q, \mathbf{z}_{ms}, t \right) \right\|_2^2 \right] \tag{10}$$

**MI2M** We add image(as noise) to the query mask $\mathbf{z}_{mq}$, $\mathbf{z}_{mq}^{(t)} = \sqrt{\bar{\alpha}_t} \mathbf{z}_{mq} + \sqrt{1 - \bar{\alpha}_t} \mathbf{z}_q$. The supervised form is as follows:

$$\mathcal{L}_{\textbf{MI2M}} = \mathbb{E}_{(\mathbf{z}_s, \mathbf{z}_q, \mathbf{z}_{ms}, \mathbf{z}_{mq}) \sim \mathcal{D}, t \in \mathcal{U}(T)} \left[ \left\| \mathbf{z}_{mq} - v_\theta^* \left( \mathbf{z}_{mq}^{(t)}, \mathbf{z}_s, \mathbf{z}_q, \mathbf{z}_{ms}, t \right) \right\|_2^2 \right] \tag{11}$$

## A.3 Cross-attention tokenized interaction

In the Section 4.2, we only discussed how to inject information from the support mask based on the Self-attention kv fusion method. Here we discuss how to inject information from the support mask based on the Tokenized Interaction Cross-Attention method. There are also the following four ways.

a. **Concatenation** We can convert the support mask $\mathbf{M}_s$ into an RGB image, encode $I_s$ and $\mathbf{M}_s$ into token sequences using CLIP image encoder respectively, concatenate them on the sequence, and finally use them as the input of cross-attention.

b. **Multiplication** We can directly multiply $\mathbf{M}_s$ on the image $\mathbf{I}_s$ to form the image $\mathbf{I}_s^* = \mathbf{I}_s \cdot \mathbf{M}_s$, and finally encode $\mathbf{I}_s^*$ into a token sequence using CLIP image encoder as the input of cross-attention.

c. **Addition** We can also directly add $\mathbf{M}_s$ to the image $\mathbf{I}_s$ to form the image $\mathbf{I}_s^* = 0.5\mathbf{I}_s + 0.5\mathbf{M}_s$. Similarly, we encode $\mathbf{I}_s^*$ into a token sequence using CLIP image encoder as the input of cross-attention.

d. **Attention Mask** We can use $\mathbf{M}_s$ as an attention mask to control self-attention, so that only $\mathbf{K}_s$ in the masked area can be accessed by $\mathbf{Q}_q$.

## A.4 Post processing

The original prediction of the model is an RGB three-channel image. We first average over the channel dimension to obtain a single-channel $\hat{\mathbf{M}}_q \in [0, 1]^{H \times W}$. Then we tried two thresholding methods, absolute threshold $\tau_a$ and relative threshold $\tau_r$. The absolute threshold is a fixed value, and the final binary mask $\mathbf{M}_q$ can be represented as:

$$\mathbf{M}_q = \begin{cases} 1, \textbf{if } \hat{\mathbf{M}}_q > \tau_a \\ 0, \textbf{otherwise} \end{cases} \tag{12}$$

Using relative threshold, we have:

$$\mathbf{M}_q = \begin{cases} 1, \textbf{if } \hat{\mathbf{M}}_q > \tau_r \max(\hat{\mathbf{M}}_q) \\ 0, \textbf{otherwise} \end{cases} \tag{13}$$

Our experiments (see Table 3) have shown that the relative threshold method achieved better results on COCO-$20^i$ [52] fold0. The optimal $\tau_r$ is 0.25.

**Table 3** – Comparison of different thresholding methods

| $\tau_r$ | 0.2 | 0.25 | 0.3 | 0.35 | 0.4 |
|---|---|---|---|---|---|
| mIoU | 47.56 | 47.69 | 47.48 | 47.4 | 47.11 |
| $\tau_a$ | 0.1 | 0.15 | 0.2 | 0.25 | 0.3 |
| mIoU | 46.64 | 47.21 | 46.91 | 46.53 | 46 |

**Table 4** – Comparison of different Multiplication methods

| Multiplication | mIoU |
|---|---|
| latent | 32.14 |
| RGB | 33.12 |

## A.5 More ablation studies

**Multiplication** We found in the experiment that Multiplication can be directly applied to RGB images, and another choice is to apply it to the latent space.

As shown in 4, the Multiplication method directly applied to RGB images achieved better results. However, the overall disparity is not significant.

**Self-Attention fusion** In previous sections, we mentioned that we use a KV fusion strategy. An alternative is to use a QKV fusion strategy, in which we also concatenate $\mathbf{Q}_q$ and $\mathbf{Q}_s$ to form $\mathbf{Q}_{qs} = [\mathbf{Q}_q, \mathbf{Q}_s]$.

This strategies means the support image can also access the query image information. As shown in

**Table 5** – Comparison of different Self-Attention fusion strategies

| strategy | mIoU |
|---|---|
| KV fusion | 46.64 |
| QKV fusion | 46.61 |

the Table 5, KV fusion is slightly better than QKV fusion, and KV fusion has lower computational complexity, which can effectively reduce memory usage and inference time. Therefore, we choose KV fusion as our default strategy.

## A.6 Other visualization

To better explore the capabilities of DiffewS, we visualize its performance on COCO-20$^i$ [52] LVIS-92$^i$ [22] and several cases from Internet. Figure 8 shows the remarkable results of DiffewS on COCO-20$^i$. Figure 6 demonstrates the impressive generalization capabilities of DiffewS. For some categories not present in the training set, such as apron and violin, DiffewS is able to perform accurate segmentation. In addition, DiffewS is demonstrated effective results in some cross-style segmentation and small object segmentation cases. For abstract concepts, such as Western dragons and Chinese dragons, DiffewS links them together to achieve accurate results. We speculate that the impressive generalization ability of DiffewS stems from its effective utilization of prior knowledge from the diffusion model. As shown in Figure 7, DiffewS also fails to segment some challenging cases. When there is a significant appearance disparity between the reference image and the target image (Appearance disparity), DiffewS may encounter segmentation errors. Additionally, if there are other objects with similar appearances in the target image (Look-alike interference) or if the objects in the image are severely occluded (Occlusion interference), DiffewS struggles to produce accurate results.

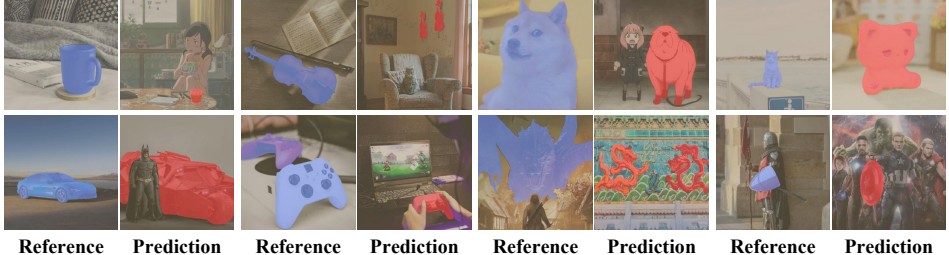

**Reference**  **Prediction**  **Reference**  **Prediction**  **Reference**  **Prediction**  **Reference**  **Prediction**

**Figure 6** – Visualization of one-shot semantic segmentation on various Internet cases. The blue color denotes the support mask while the red represents the query mask. DiffewS also performs impressively on cases with cross-styles and significant appearance differences, as well as on abstract concepts it has never encountered before.

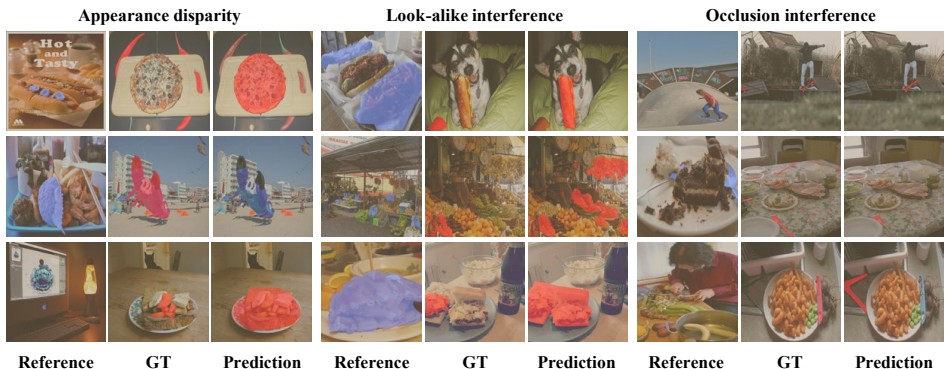

**Reference**  **GT**  **Prediction**  **Reference**  **GT**  **Prediction**  **Reference**  **GT**  **Prediction**

**Figure 7** – Three types of failed cases in one-shot semantic segmentation on LVIS-92$^i$ and COCO-20$^i$.

## A.7 N-shot studies

As mentioned in Section 4.5, we conduct Inference Time Improvement on COCO and PASCAL, Training Time Improvement on COCO fold-0, see Table 6 and Table 7.

**Table 6** – Performance of DiffewS with Inference Time Improvement

|                  | 1-shot | 5-shot | 10-shot |
|------------------|--------|--------|---------|
| **COCO**         |        |        |         |
| Diffews (ori)    | 71.3   | 72.2   | 70.1    |
| Diffews (sample) | 71.3   | 74.7   | 73.4    |
| **PASCAL**       |        |        |         |
| Diffews (ori)    | 88.3   | 87.8   | 87.2    |
| Diffews (sample) | 88.3   | 89.4   | 89.6    |

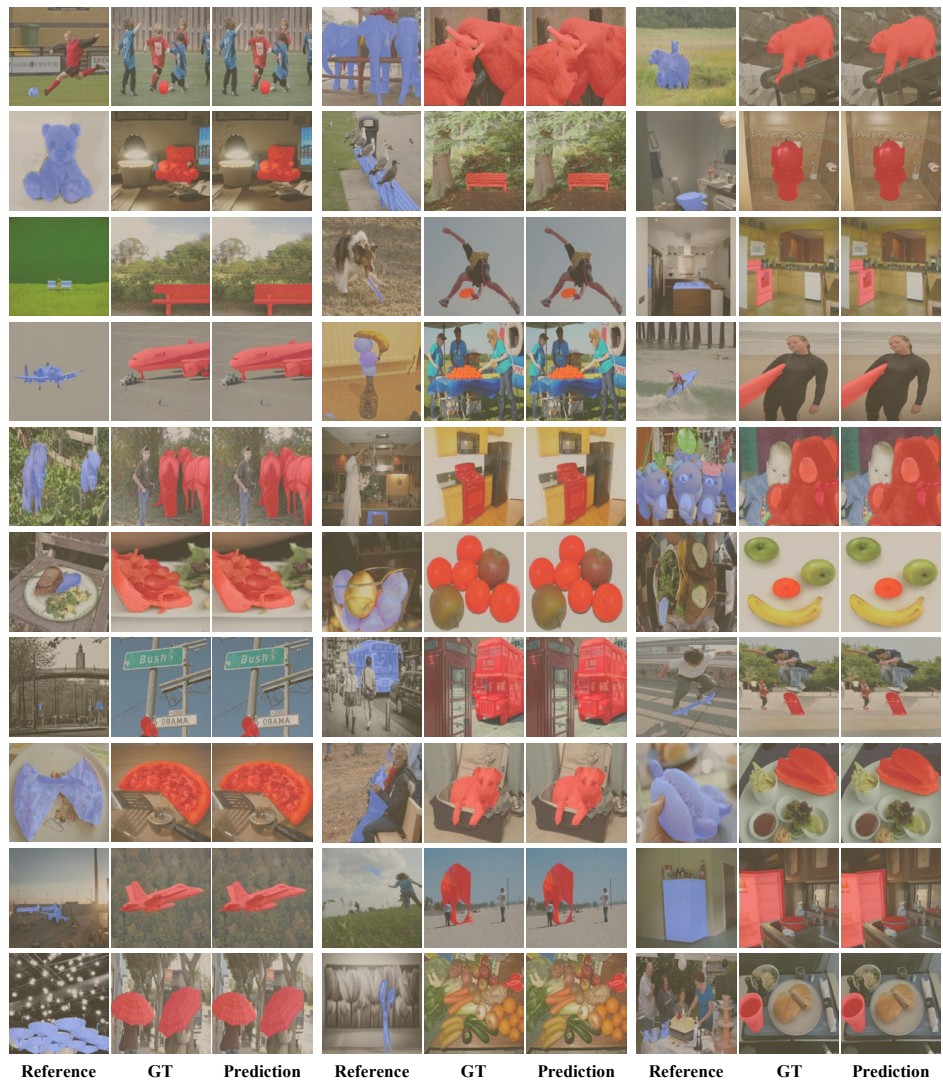

| Reference | GT | Prediction | Reference | GT | Prediction | Reference | GT | Prediction |

**Figure 8** – Qualitative results of one-shot semantic segmentation on COCO-20$^i$. The blue color denotes the support mask while the red represents the query mask. DiffewS has an impressive performance on COCO-20$^i$.

**Table 7** – Performance of DiffewS with Training Time Improvement

|  | 1-shot | 5-shot | 10-shot |
|---|---|---|---|
| Diffews (ori, train 1 shot) | 47.7 | 52.0 | 49.1 |
| Diffews (train 1-5 shot) | 46.4 | 57.6 | 55.9 |
| Diffews (train 1-7 shot) | 47.1 | 57.3 | 58.7 |

