# OpenReview forum: "Unleashing the Potential of the Diffusion Model in Few-shot Semantic Segmentation"
_NeurIPS.cc/2024/Conference — NeurIPS 2024 poster_

### Official Review · Reviewer_zc26 · 2024-07-06

**Soundness:** 4
**Presentation:** 4
**Contribution:** 4
**Rating:** 7
**Confidence:** 4

**Summary:**

This paper explores the potential of the Latent Diffusion Model for Few-shot Semantic Segmentation. The authors study four crucial elements of applying the Diffusion Model to Few-shot Semantic Segmentation and propose several reasonable solutions for each aspect.
Based on their observations, the authors establish the DiffewS framework, which maximally retains the generative framework and effectively utilizes the pre-training prior.
Experimental results demonstrate that the proposed method outperforms the previous SOTA models in multiple settings.

**Strengths:**

- The topic is interesting and the proposed method is novel. How to better utilize the diffusion model for perception tasks is currently a promising direction.
And this paper takes a first step to apply diffusion model to few-shot semantic segmentation.
- The writing is clear and easy to follow.   The author presents clear and logical thinking, starting from four key elements of implementing FSS, proposing a series of reasonable solutions, and validating the effectiveness of these solutions through experiments. The KV-fusion strategy, the supervisory form of the query mask, or the single-step generation strategy, all of these provide certain insights for subsequent research on FSS and visual perception.
- The Diffews method has demonstrated strong performance in in-context settings, especially in in-domain scenarios. It also shows excellent performance in strict few-shot settings and converges faster compared to traditional methods.

**Weaknesses:**

- Some places lack proper references. For example, in L110, "v-prediction," we typically cite Tim Salimans and Jonathan Ho's work: "Progressive distillation for fast sampling of diffusion models" (ICLR, 2022).
In L241, I noticed that the appendix contains an analysis of the training objectives for different generation processes, and a reference should be added here.

- In Section 4.4, L257, the variances $\beta_1$ and $\beta_2$  are not clearly defined. Since $\beta_t$ in L106 represents the variance at time t, it would be better to use $\beta^1$ and $\beta^2$  to denote these two different settings here. Additionally, the meaning of the two values within the tuple is not explained. In my view,  $\beta^1 = (\beta^{start},\beta^{end})$ should represent the initial and final values of the DDIM scheduler's $\beta$ .
Furthermore, it is necessary to explain the intuitive differences between these two settings.

- Other typos:
L273,  "we tested" -> "we test"
L151, "linear projection layer. ." ->"linear projection layer"
L279, "function as"-> "functions as"

**Questions:**

I've noticed that the current method doesn't perform well enough on 5-shot, can this be mitigated by making adjustments during training?

**Limitations:**

NaN

---

> ### Author Rebuttal · Authors · 2024-08-04
>
> We deeply appreciate your acknowledgment of our motivation and approach. We will try our best to address the questions and weaknesses you have raised.
>
> **W1: Lack of proper references**
> Thank you for your feedback. We will add appropriate references in the next version.
>
> **W2: Definition of $\beta_{1}$ and $\beta_{2}$**
> Yes, you are correct. We will clearly define $\beta^{1}$ and $\beta^{2}$ in Section 4.4 and explain the intuitive meanings of the two values. $\beta^{1}=(\beta_{start},\beta_{end})$, where $\beta_{start}$ and $\beta_{end}$ respectively represent the initial and final values of $\beta$ in the DDIM scheduler. We also mentioned this in our discussion with Reviewer uveh. Intuitively, $\beta^{2}$ adds more noise at smaller time steps compared to $\beta^{1}$. Experimental results show that adding more noise at smaller time steps improves the model's performance, which is why we ultimately chose OI2M. .
>
> **W3: Other typos**
> Thank you once again. We will fix these errors in the next version.
>
> **Questions: 5-shot performance**
>
> We are grateful for your question. We have also mentioned this issue in our response to Reviewer zc26. We have tried two approaches to address the N-shot setting issue.
>
> **Inference Time Improvement**
>
> The first approach is an improvement in the inference phase. Considering the transition from 1-shot to n-shot, where additional support samples' Keys and Values are concatenated (see Appendix A.6), this leads to a significantly larger number of Keys and Values for the self-attention layer to process during inference compared to the training phase. An intuitive solution is to randomly sample the Keys and Values of the support samples at each layer during inference, ensuring the number of samples matches the number of Keys and Values during training. We found that this approach indeed improves the model's performance in 5-shot and 10-shot settings. The experimental results are shown in the table below.
>
> Table: Performance of DiffewS with Inference Time Improvement
> |                     | 1-shot | 5-shot | 10-shot |
> |---------------------|--------|--------|---------|
> | **COCO**            |        |        |         |
> | Diffews (ori)       | 71.3   | 72.2   | 70.1    |
> | Diffews (sample)    | 71.3   | 74.7   | 73.4    |
> | **PASCAL**          |        |        |         |
> | Diffews (ori)       | 88.3   | 87.8   | 87.2 |
> | Diffews (sample)    | 88.3   | 89.4   | 89.6 |
>
> Another more straightforward idea is to introduce multiple support samples during the training phase. This way, the model can learn how to select among multiple support images during training. To save computational resources, we conducted experiments under the ablation study setting on the COCO fold0. The experimental results are shown in the table below.
>
> Table: Performance of DiffewS with Training Time Improvement
> |                           | 1-shot | 5-shot | 10-shot |
> |---------------------------|--------|--------|---------|
> | Diffews (ori, train 1 shot)| 47.7   | 52.0   | 49.1    |
> | Diffews (train 1-5 shot)   | 46.4   | 57.6   | 55.9    |
> | Diffews (train 1-7 shot)   | 47.1   | 57.3   | 58.7    |
>
> (train 1-5 shot) indicates that we randomly select 1 to 5 support samples as input during a single training iteration. From the table, we can see that when multiple support samples are introduced during the training phase, the model's performance significantly improves in both 5-shot and 10-shot scenarios. Additionally, as the number of support samples increases during training, the model's performance in the 10-shot scenario gradually improves. We also found that when multiple support samples are introduced during the training phase, the 1-shot performance may decrease. This could be due to the inconsistency between training and inference.
>
> In summary, both approaches effectively alleviate the issues associated with the N-shot setting, demonstrating the potential of DiffewS in Few-shot Semantic Segmentation. We also hope that our work can provide some inspiration for researchers in the fields of diffusion-based models and Few-shot Semantic Segmentation.

---

> > ### Comment · Reviewer_zc26 · 2024-08-11
> >
> > I would like to thank the authors for their detailed responses. All my concerns were thoroughly addressed. Overall,  this work provides clear insights and significant contributions to the field. Therefore, I will maintain my original rating.

---

> > > ### Author Response · Authors · 2024-08-12
> > >
> > > Thank you very much for your thoughtful review and for acknowledging our work. We are delighted that our responses addressed your concerns, and we greatly appreciate your recognition of the insights and contributions our work offers to the field. Your positive feedback is invaluable to us.
> > >
> > > We also want to assure you that we will incorporate your feedback and any potential improvements into future versions of our paper.

---

### Official Review · Reviewer_uveh · 2024-07-10

**Soundness:** 3
**Presentation:** 3
**Contribution:** 2
**Rating:** 6
**Confidence:** 5

**Summary:**

This paper explores the potential of diffusion model in Few-Shot Segmentation (FSS) tasks. For achieve better performance, it examines four critical elements of applying the diffusion model to FSS. Building on this research, the paper introduces the DiffewS framework, with experimental results validating its effectiveness. The authors assert that this is the first diffusion-based model specifically designed for Few-shot Semantic Segmentation.

**Strengths:**

This article comprehensively explores four key aspects of adapting the Diffusion Model to FSS tasks: the interaction between query and support images, injection of support mask information, supervision from query masks, and exploration of the generation process. Multiple implementation approaches regarding the above four aspects are studied and compared through experiments. And the relevant experimental results, presented in figures 2, 3, and 4, are convincing.

**Weaknesses:**

The weakness of this article is as follows:

1. Why should the diffusion model be applied to FSS tasks? What advantages does DM offer over Multi-modal fundamental models (like SAM or LLaVA) for FSS tasks? This article's research motivation requires stronger support through theoretical explanations or experimental evidence.
2. The default generation process (OI2M) transforms "gradually approaching ground truth in denoising" to "single-step prediction in segmentation", yielding optimal performance. Does this suggest that DM's multi-step denoising process is unsuitable for FSS? Does this paper not unleash the potential of DM, instead transforming it into a segmentation model? Section 4.4 should include more theoretical explanations.
3. The comparison of inference speed and training cost is important and should be supplemented to illustrate the advantages of DiffewS.
4. Other studies that apply the Diffusion Model to Few-Shot Segmentation (e.g., DifFSS[1], Ref LDM-Seg[2]) need to be compared.

[1] DifFSS: Diffusion Model for Few-Shot Semantic Segmentation. Github: https://github.com/donglongzi/DifFSS-BAM/tree/main

[2] Explore In-Context Segmentation via Latent Diffusion Models. Project Page: https://wang-chaoyang.github.io/project/refldmseg/

**Questions:**

1. Did the pretrained Stable Diffusion (SD) weights utilized in the model training? How much parameters were updated?
2. What are the experimental settings of the comparing results in Fig2/3/4?
3. In Section 4.4, this paper chose OI2M as the generation pipeline. Since it is a one-step process, what is the timestep setting in this pipeline during training? And did any accelerated generation schedule (such as DDIM) is utilized?
4. Did the few-shot support images being processed as the same as one-shot support image in support image encoding branch? What is the difference of model settings between using few-shot support images and one-shot support image? How about the comparison of computational cost?
5. In Section 5.2, the “In-context” setting means “these specialist models are also trained on the test categories from COCO and PASCAL VOC”. Is it a general experimental dataset setting? And is it a fair comparison for the methods in Table 1?

---

> ### Author Rebuttal · Authors · 2024-08-04
>
> We sincerely thank you for your very careful and detailed review. We are delighted that you found our work to be comprehensive and convincing. The questions and weaknesses you've pointed out are incredibly helpful to us, and we will do our utmost to address them below.
>
> **W1: Why should the diffusion model be applied to FSS tasks?**
> In L43-L53, we briefly introduce the motivation for applying the diffusion model to FSS tasks. We will make this clearer in the revised version.  The  pretrained diffusion model exhibits significant potential in fine-grained pixel prediction ability[1,2]
> and semantic correspondence ability [3-5]. Both of these capabilities are essential for Few-Shot Segmentation (FSS) tasks, making diffusion models particularly suitable for this application.
>
> For example, SAM has strong fine-grained pixel prediction capabilities but lacks semantic abilities, leading to a series of works attempting to address this issue[6,7]. Therefore, it is difficult to use SAM in FSS. For example, PerSAM[8] is based solely on SAM's capabilities but does not perform well in FSS.
>
> On the other hand, LLaVA has strong semantic abilities but lacks fine-grained pixel prediction capabilities due to its reliance on only image-text pairs for pretraining(lacks fine-grained supervision). Therefore, LISA[7] needs to rely on SAM to achieve segmentation prediction.
>
>
>
> **W2: Multi-step denoising process**
> We believe that the supervised objective of the multi-step denoising process is not suitable for FSS, but this does not mean that single-step prediction fails to unleash the potential of the diffusion model (DM). The difference between OI2M and MI2M/MN2M lies only in their training objectives (see Appendix A.2). Compared to OI2M, MI2M and MN2M use noisy target masks as input, which could potentially lead to information leakage in segmentation tasks, especially when the noise level is low, thus hindering the model's learning. The experimental results in Figure 3 also confirm this, showing that setting $\beta_{2}$ is more effective than $\beta_{1}$. Intuitively, $\beta_{2}$ adds more noise at smaller time steps compared to $\beta_{1}$, and OI2M can be understood as an extreme case where the input contains no information about the target mask at all (or the target mask has become pure noise). We will provide more theoretical explanations in Section 4.4.
>
> **W3: Inference speed and training cost**
> Thank you for your suggestions. We can compare the advantages of DiffewS in two aspects:
>
> 1.Compared to traditional FSS methods, DiffewS converges faster and has lower training overhead. For instance, taking DCAMA[9] as an example, both our model and DCAMA were trained on 4 V100 GPUs. The comparison of training time is as follows:
>
> Table: Training Time Comparison
> | Method  | Training Time |
> |---------|---------------|
> | DCAMA   | 36h           |
> | DiffewS | 6h            |
>
> 2. Compared to other generative methods based on multi-step denoising, DiffewS has a faster inference speed. For instance, considering MI2M and MN2M mentioned in this paper, the inference speed comparison is as follows:
>
> Table: Inference Speed Comparison
> | Method  | Forward times  |
> |---------|-----------------|
> | OI2M    | x1              |
> | MI2M    | x10             |
> | MN2M    | x10             |
>
> In our implementation, MI2M and MN2M use DDIM sampling, requiring the model to forward 10 times, while OI2M only needs to forward once. Thus, OI2M should be 10 times faster than MI2M/MN2M."
>
> **W4:Other studies**
> Thank you for your valuable suggestions. We will include discussions of DifFSS and Ref LDM-Seg in the Related Work section, as well as comparisons with our work. DifFSS uses generated images as auxiliary support images, which can improve the performance of existing FSS methods to some extent. This approach enhances FSS performance from the data level, making it orthogonal to our method. Ref LDM-Seg is a contemporaneous work to ours. Unlike DiffewS, which utilizes self-attention to achieve interaction between the query and support images, Ref LDM-Seg uses cross-attention for this purpose. Additionally, Ref LDM-Seg introduces multiple linear projection layers, increasing the number of parameters in the diffusion model, which might also disrupt the original priors of the diffusion model. Our work provides a more comprehensive and systematic analysis of applying diffusion models to Few-shot Semantic Segmentation tasks. Considering that Ref LDM-Seg has not been open-sourced, and the training data and experimental setups cannot be strictly aligned, we currently cannot conduct a direct and fair experimental comparison.
>
> **References**
>
> [1]Ke B, Obukhov A, Huang S, et al. Repurposing diffusion-based image generators for monocular depth estimation. CVPR2024
>
> [2]Lee H Y, Tseng H Y, Yang M H. Exploiting Diffusion Prior for Generalizable Dense Prediction. CVPR2024
>
> [3]Tang, Luming, et al. Emergent correspondence from image diffusion. NeurIPS2023
>
> [4]Luo, Grace, et al. Diffusion hyperfeatures: Searching through time and space for semantic correspondence. NeurIPS2023
>
> [5]Zhang, Junyi, et al. A tale of two features: Stable diffusion complements dino for zero-shot semantic correspondence. NeurIPS2023
>
> [6]Li F, Zhang H, Sun P, et al. Semantic-sam: Segment and recognize anything at any granularity. ECCV 2024
>
> [7]Lai X, Tian Z, Chen Y, et al. Lisa: Reasoning segmentation via large language model. CVPR 2024
>
> [8]Zhang R, Jiang Z, Guo Z, et al. Personalize segment anything model with one shot. ICLR2024
>
> [9]Shi, Xinyu, et al. Dense cross-query-and-support attention weighted mask aggregation for few-shot segmentation. European Conference on Computer Vision.  ECCV2022
>
>
> **Due to space constraints, we will respond to your valuable questions in the global response.**

---

> > ### Comment · Reviewer_uveh · 2024-08-13
> >
> > Thanks to the author's thorough response, most of my concerns have been addressed. Therefore, I am upholding the original rating.

---

> > > ### Author Response · Authors · 2024-08-13
> > >
> > > Thanks again for your thoughtful review and for your recognition of our efforts to address your concerns. We deeply appreciate your professional insights and the constructive feedback that has helped us improve our work.

---

### Official Review · Reviewer_WLLp · 2024-07-13

**Soundness:** 4
**Presentation:** 4
**Contribution:** 3
**Rating:** 6
**Confidence:** 4

**Summary:**

This paper introduces DiffewS, a novel Diffusion-based generalist model designed for few-shot segmentation. It systematically examines four key components involved in applying Diffusion Models to Few-shot Semantic Segmentation. For each component, the work proposes several viable solutions, which are validated through extensive experiments. Notably, the work introduces Key-Value Fusion Self-Attention (FSA) to facilitate the interaction between the query and support images.

**Strengths:**

1. The idea is clear and readily comprehensible
2. The writing is of high quality and the structure is coherent.
3. The experiments are adequate and demonstrate the eff ectiveness of the framework.

**Weaknesses:**

The method of incorporating support masks, adapted from Marigold, may have limitations in a few-shot setting. The experiments detailed in Table 1 indicate that DiffewS achieves only marginal improvements in the COCO few-shot scenario compared to the one-shot scenario and even exhibits a decline in performance on the PASCAL dataset. Further evaluation of DiffewS’s capabilities with additional support samples, such as through experiments in a ten-shot setting, would be beneficial.

**Questions:**

I am highly intrigued by the possibility of whether DiffewS can perform open-vocabulary segmentation by substituting the support mask with text.

**Limitations:**

The limitation is discussed in the paper supplementary material.

---

> ### Author Rebuttal · Authors · 2024-08-04
>
> Thank you very much for acknowledging our motivation and approach. The questions and weaknesses you've pointed out are incredibly helpful to us, and we will do our utmost to address them below.
>
> **Weaknesses: N-shot setting**
>
> Thanks again for pointing out this issue. As we mentioned in the Experiment and Limitation sections of our paper, our current work primarily focuses on unleashing the potential of Diffusion Models in Few-shot Semantic Segmentation. Consequently, our experiments and optimizations are mainly concentrated on the One-shot setting. Since the model receives only one support image at a time during the training phase, using 5-shot or 10-shot during the inference phase can indeed cause inconsistencies. The model might not learn how to effectively utilize multiple support images during training, which could result in interference from lower-quality samples. However, we fully agree with your point that we should further explore the performance of DiffewS with additional support samples. Reviewer zc26 raised a similar request, asking, "can this be mitigated by making adjustments during training?"
>
> Therefore, we have tried two approaches to address the N-shot setting issue.
>
> **Inference Time Improvement**
>
> The first approach is an improvement in the inference phase. Considering the transition from 1-shot to n-shot, where additional support samples' Keys and Values are concatenated (see Appendix A.6), this leads to a significantly larger number of Keys and Values for the self-attention layer to process during inference compared to the training phase. An intuitive solution is to randomly sample the Keys and Values of the support samples at each layer during inference, ensuring the number of samples matches the number of Keys and Values during training. We found that this approach indeed improves the model's performance in 5-shot and 10-shot settings. The experimental results are shown in the table below.
>
>
>
> Table: Performance of DiffewS with Inference Time Improvement
> |                     | 1-shot | 5-shot | 10-shot |
> |---------------------|--------|--------|---------|
> | **COCO**            |        |        |         |
> | Diffews (ori)       | 71.3   | 72.2   | 70.1    |
> | Diffews (sample)    | 71.3   | 74.7   | 73.4    |
> | **PASCAL**          |        |        |         |
> | Diffews (ori)       | 88.3   | 87.8   | 87.2 |
> | Diffews (sample)    | 88.3   | 89.4   | 89.6 |
>
>
> From the table, we can see that following the original DiffewS inference method, the 10-shot performance is even lower than the 1-shot result. This might be due to the reasons we analyzed earlier. However, after adopting the random sampling strategy during inference, there is a significant improvement in the model's performance for both 5-shot and 10-shot scenarios
>
> **Training Time Improvement**
>
> Another more straightforward idea is to introduce multiple support samples during the training phase. This way, the model can learn how to select among multiple support images during training. To save computational resources, we conducted experiments under the ablation study setting on the COCO fold0. The experimental results are shown in the table below.
>
> Table: Performance of DiffewS with Training Time Improvement
> |                           | 1-shot | 5-shot | 10-shot |
> |---------------------------|--------|--------|---------|
> | Diffews (ori, train 1 shot)| 47.7   | 52.0   | 49.1    |
> | Diffews (train 1-5 shot)   | 46.4   | 57.6   | 55.9    |
> | Diffews (train 1-7 shot)   | 47.1   | 57.3   | 58.7    |
>
> (train 1-5 shot) indicates that we randomly select 1 to 5 support samples as input during a single training iteration. From the table, we can see that when multiple support samples are introduced during the training phase, the model's performance significantly improves in both 5-shot and 10-shot scenarios. Additionally, as the number of support samples increases during training, the model's performance in the 10-shot scenario gradually improves. We also found that when multiple support samples are introduced during the training phase, the 1-shot performance may decrease. This could be due to the inconsistency between training and inference.
>
> In summary, both approaches effectively alleviate the issues associated with the N-shot setting, demonstrating the potential of DiffewS in Few-shot Semantic Segmentation. We also hope that our work can provide more inspiration for researchers in the fields of diffusion-based models and Few-shot Semantic Segmentation.
>
> **Questions:open-vocabulary segmentation**
>
> We believe it is feasible; however, it would be more appropriate to inject text information through cross-attention, as discussed in (Sec 4.1 Tokenized Interaction Cross-Attention). A similar idea has already been validated in the Referring Segmentation task (UniGS [1]). We will include a discussion on this in the Related Work section. Our work focuses more on how to effectively achieve interaction between the features of two images, which is why we ultimately chose the KV Fusion Self-Attention approach. However, these two methods are not mutually exclusive and can coexist. Nevertheless, to achieve open-vocabulary segmentation, more segmentation datasets with richly annotated text are needed for training, as merely using a few-shot dataset is insufficient. Therefore, we have not conducted experimental verification.
>
> [1] Qi L, Yang L, Guo W, et al. Unigs: Unified representation for image generation and segmentation[C]//Proceedings of the IEEE/CVF Conference on Computer Vision and Pattern Recognition. 2024: 6305-6315.

---

> > ### Comment · Reviewer_WLLp · 2024-08-13
> >
> > Thank the authors for the detailed rebuttal, which has satisfactorily addressed my concerns. I believe that the paper has merit and should be accepted. I will be keeping my original rating.

---

> > > ### Author Response · Authors · 2024-08-14
> > >
> > > We sincerely appreciate your positive feedback and are grateful for your recognition of the merits of our work. Your constructive comments have been invaluable in enhancing the quality of our paper.
> > >
> > > Thank you again for your time and expertise.

---

### Author Rebuttal · Authors · 2024-08-04

Dear Reviewers,

We would like to extend our heartfelt thanks to all of you for taking the time and demonstrating professionalism in thoroughly evaluating our work. Your constructive feedback has been immensely helpful in further improving the quality of our work and refining our research. We are also greatly encouraged by your recognition of our efforts.

We will carefully address the issues and suggestions raised by the reviewers and will make further revisions and improvements to our paper. Below each Official Review, we have provided responses to the questions and suggestions made by the reviewers, and we hope these responses adequately address the issues and concerns raised.

Due to the character limit for each review's rebuttal, we have moved some of the responses to Reviewer **uveh**'s questions here.

**Q1: Pretrained Stable Diffusion (SD) weights**
Yes, we utilized the pretrained Stable Diffusion (SD) weights in our model training. We keep the VAE fixed and only fine-tune the Unet.

**Q2: Experimental settings of Fig2/3/4**
The experimental settings for the results in Fig2/3/4 are our ablation study settings. We will provide more detailed experimental settings in the revised version. Specifically, We use one 4090 GPU to train the model with a batch size of 4, other settings are strictly aligned with the Table 2 settings. There are small differences between Fig3 and Fig4(46.64 vs 47.7).
This is because we did not use an optimized threshold in Fig. 3. In Fig. 3, we used different forms of mask supervision, each maybe corresponding to a different optimal threshold. See Appendix A.4 for more details.

**Q3: OI2M timestep setting**
We set the timestep to 1 in the OI2M pipeline during training. We don't use any ccelerated generation schedule like DDIM, because it is a one-step process. We also add an additional ablation study to compare different timestep settings(the input time embbeding of unet) during training, the results are shown in Table below.

Tabel: Performance of DiffewS with different timestep settings
| Timestep | mIoU  |
|----------|-------|
| 1        | 47.7  |
| 10       | 47.8  |
| 50       | 47.2  |
| 100      | 43.14 |

It can be observed that the performance is better with smaller timesteps. This is because the UNet in the original diffusion model receives noiseless real images as input at smaller timesteps, and our current model also receives noiseless real images as input.

**Q4: Few-shot support images**
Yes, the few-shot support images are processed in the same way as the one-shot support image in the support image encoding branch. We use the same model weights for both one-shot and few-shot settings. The only difference is that we concatenate the Key and Value of the support images during self-attention(see Appendix A.6).
Since our model can process multiple support images in batches, the inference time doesn't increase significantly. Specifically, when evaluating on COCO fold0, the inference time for 5-shot is approximately twice that of 1-shot, and the GPU memory usage is also about twice that of 1-shot.


**Q5: In-context setting**
Yes, the "In-context" setting is a general experimental dataset setting.
This setting was first proposed by SegGPT[A] and has been adopted by works such as Matcher[B] and PerSAM[C]. The comparison is fair, as these specialist models were also trained on COCO and PASCAL VOC test categories. The specific results can be found in Table 1 of the SegGPT paper. We will provide a more detailed description in the experimental section.

[A] SegGPT: Segmenting Everything In Context ICCV2023

[B] Matcher: Segment Anything with One Shot Using All-Purpose Feature Matching ICLR2024

[C] Personalize Segment Anything Model with One Shot ICLR2024

---

### Decision · Program_Chairs · 2024-09-25

**Decision:**

Accept (poster)

**Comment:**

The paper introduces a latent diffusion model (DM)-based method for few-shot segmentation (FSS). It develops several effective solutions from four key aspects of applying DM to FSS and achieves strong performance on several benchmarks.

The paper received mostly positive reviews. The three reviewers considered the proposed approach novel and effective (WLLP, zc26), the paper well-written (WLLP, zc26), the study of its solutions comprehensive (WLLP, uveh) and the experimental results strong/convincing (WLLP, uveh, zc26). However, they initially also raised several important concerns regarding its mixed performance under multi-shot settings (WLLP, uveh, zc26), insufficient motivation (uveh), and the lack of clarity in technical details and settings (uveh, zc26). The author's rebuttal provided thorough clarifications on its motivation and technical details, and additional results on 5 and 10-shot settings, which largely addressed the reviewers' concerns. During the rebuttal discussion, the three reviewers reached a consensus with positive recommendations on this work.

After considering the paper, the accompanying reviews, and the rebuttal discussion, despite some limitations in the multi-shot setting, the AC concurs with the reviewers regarding the substantive contributions of this work and therefore recommends it for acceptance. The author should take into account the rebuttal and the reviewers' feedback, especially regarding the refined 5 and 10-shot results, when preparing the final version.